# Ultrafast Antisolvent Growth of Single-Crystal CsPbBr_3_ Microcavity for Whispering-Gallery-Mode Lasing

**DOI:** 10.3390/nano13142116

**Published:** 2023-07-20

**Authors:** Li Zhang, Xinxin Li, Yimeng Song, Bingsuo Zou

**Affiliations:** 1School of Materials Science and Engineering, Beijing Institute of Technology, Beijing 100081, China; 3120195600@bit.edu.cn; 2Key Laboratory for Renewable Energy, Beijing Key Laboratory for New Energy Materials and Devices, Beijing National Laboratory for Condensed Matter Physics, Institute of Physics, Chinese Academy of Sciences, Beijing 100190, China; xinxin.li@iphy.ac.cn; 3Guangdong Provincial Key Laboratory of Electronic Functional Materials and Devices, Huizhou University, Huizhou 516001, China; b20180340@xs.ustb.edu.cn; 4State Key Laboratory of Featured Metal Materials and Life-Cycle Safety for Composite Structures, Guangxi Key Laboratory of Processing for Nonferrous Metals and Featured Materials, School of Resources, Environments and Materials, Guangxi University, Nanning 530004, China

**Keywords:** perovskites, fast growth, microstructures, lasing, photonics

## Abstract

In recent years, all-inorganic cesium lead bromide (CsPbBr_3_) perovskites have garnered considerable attention for their prospective applications in green photonics and optoelectronic devices. However, the development of efficient and economical methods to obtain high-quality micron-sized single-crystalline CsPbBr_3_ microplatelets (MPs) has become a challenge. Here, we report the synthesis of CsPbBr_3_ MPs on Si/SiO_2_ substrate by optimizing the ultrafast antisolvent method (FAS). This technique is able to produce well-dispersed, uniformly sized, and morphologically regular tetragonal phase single crystals, which can give strong green emission at room temperature, with excellent stability and excitonic character. Moreover, the crystals demonstrated lasing with a whispering gallery mode with a low threshold. These results suggest that the single-crystalline CsPbBr_3_ MPs synthesized by this method are of high optical quality, holding vast potential for future applications in photonic devices.

## 1. Introduction

Small lasers show great application potential in optical integration, high-speed communication, and high-resolution imaging. Over the past few decades, a wide variety of semiconductor nanostructures, including nanowires, nanoribbons, and quantum dots, have been used as active materials for small lasers [1,2,3,4,5,6]. Recently, lead halide perovskites have attracted considerable interest in photovoltaics research due to their high absorption coefficient, substantial carrier diffusion length, and minimal density of defects. These properties make them highly suitable for efficient light absorption and carrier transport, which are crucial for photovoltaic applications. Additionally, perovskites exhibit superior gain performance, making them excellent candidates for coherent light sources [7,8,9]. Their superior gain performance also positions them as key contenders for efficient coherent light sources. Perovskite-based miniaturized lasers have been successfully developed utilizing various micro/nanostructures like nanowires, nanoplatelets, and quantum dots, all operating efficiently at room temperature [10,11,12,13]. When compared with traditional II-VI/III-V semiconductors, these optically pumped perovskite micro-lasers have demonstrated impressive advantages such as a lower lasing threshold, improved spectral coherence, and a broad spectrum of emission colours.

In recent years, all-inorganic perovskites, particularly CsPbX_3_ (where X = Cl, Br, and I), have emerged as promising materials for various optoelectronic applications. One of the main reasons for their growing preference over hybrid organic–inorganic perovskites is their superior electronic properties and enhanced stability against environmental factors such as moisture, oxygen, and heat [14,15,16]. As a result, these materials are particularly favoured for laser gain materials, which hold promising applications in areas like on-chip optical communication, sensors, and computing [17,18,19]. Over the past few years, there have been significant advancements in the field, with amplified spontaneous emission and lasing action being reported in CsPbX_3_ thin films [20,21]. More recently, lasing action has been observed in CsPbX_3_ micro/nanostructures, including micro/nanowires, nanorods, nanoplatelets, and microspheres [22,23]. To date, both Fabry-Pérot (F-P) and whispering-gallery-mode (WGM) lasers have been achieved in these CsPbX_3_ microcavities [11,24]. Notably, WGM cavities offer advantages over F-P cavities due to their high-quality factor and compact size, which contribute to lasers with narrow linewidth and low thresholds.

At present, the main methods for synthesizing microcrystal CsPbBr_3_ MPs are an antisolvent method, a space-confined growth strategy, and templated assembly [25,26,27]. Previously, we were successful in fabricating microcrystal CsPbCl_3_ through an optimized ultrafast antisolvent method (FAS), and the mechanism is analysed deeply [28]. However, the application of this technique to other perovskites has been hindered due to their excessive solubility in DMF and the relatively constrained space during their crystallization process, which includes nucleation and growth phases. These issues result in significant interferences, causing the perovskite crystals to stack and ultimately deteriorating the quality of individual CsPbBr_3_ MPs. Hence, it is of paramount importance to enhance the ultrafast antisolvent method to overcome these challenges.

Herein, to solve this issue, we added chlorobenzene (CB) to the precursor solution to reduce the solubility of perovskite and successfully prepared a highly dispersed CsPbBr_3_ MP. The single crystal CsPbBr_3_ MPs are evenly dispersed on the substrate, and the size ranges from 8 to 25 μm, presenting a regular geometric appearance. Strong photoluminescence (PL) spectroscopy is observed at 78–298 K, which is ascribed to the recombination of excitons. Moreover, the precisely defined geometry and smooth surface facilitate the operation of WGM lasing at room temperature. The CsPbBr_3_ MP shows excellent stability under laser pulse and ambient air. These findings suggest that CsPbBr_3_ MPs, synthesized using the FAS, hold substantial promise for applications in the realms of integrated photonics and optoelectronics.

## 2. Materials and Methods

### 2.1. Materials

CsBr and PbBr_2_ were purchased from Aladdin (Shanghai, China). Dimethyl formamide (DMF), dichloromethane, chlorobenzene, ethanol, and acetone were purchased from Macklin (Shanghai, China).

### 2.2. Instrumentation

The surface structure of the synthesized samples was examined using a scanning electron microscope (SEM) via the Hitachi Instruments SU5000 (Japan), operating at a voltage of 20 kV. This was paired with energy-dispersive spectrometer (EDS) mapping to visualize the spatial distribution of various elements. Additionally, an atomic force microscope (AFM) from Bruker (Multimode8, Germany) was used to analyse surface roughness and particle thickness, operating in a scan analysis mode. The crystalline structure of the microparticles (MPs) was determined by acquiring X-ray diffraction (XRD) spectra, utilizing a PANalytical Empyrean XRD instrument (Netherlands). Transmission electron microscopy (TEM) analyses were carried out at 200 KV using a JEOL-2010F field-emission type high-resolution TEM (Japan).

X-ray photoelectron spectroscopy (XPS) was used to measure core level and VB spectra with the assistance of a Thermo Fisher Scientific ESCALAB 250X (Thermo Fisher Scientific, Waltham, MA, USA) equipped with a spherical quartz monochromator and energy analyse working in the range of binding energies from 0 to 1500 eV. The energy resolution was ΔE ≤ 0.5 eV. The samples were measured at a pressure of 10^−7^ Pa.

The photoluminescence (PL) and lasing spectra were evaluated using a custom-built confocal micro-spectrometer setup. A solid-state laser with a continuous wavelength of 405 nm was employed for the steady-state PL tests, with the laser spot being concentrated onto the samples via a 100× Olympus objective lens (numerical aperture: 0.95, Japan). The PL spectra were gathered using the same lens in a backscattering configuration, then analysed through a monochromator fitted with a charge-coupled device detector (Japan) cooled by liquid nitrogen. For experiments conducted at lower temperatures, the samples were affixed within a liquid-flow micro-cryostat (CFM-1738-102, CRYO Industries, Manchester, NH, USA). Conducting lasing measurements, a 400 nm femtosecond pulsed laser, generated by the Coherent Astrella regenerative amplifier (800 nm, 100 fs, 1 kHz, USA), served as the excitation source. The laser spot was honed onto the samples with the aid of a 50× Olympus objective lens (numerical aperture: 0.45), producing a spot with a diameter of approximately 30 µm.

### 2.3. Synthesis Method

Figure 1 shows the schematic diagram of CsPbBr_3_ MPs synthesis by FAS. Initially, 1 mmol each of CsBr and PbBr_2_ were measured. These were mixed into 1 mL of DMF, and the solution was stirred at a temperature of 80 °C for 12 h. Then, the precursor was filtered, resulting in a clear solution. To this solution, 20 μL CB was added in order to adjust the solution to its saturation. After allowing the solution to settle, the resulting clear filtrate was then utilized as the precursor for CsPbBr_3_. The Si/SiO_2_ substrate (1.5 cm × 1.5 cm) was meticulously cleaned using a series of solvents, beginning with acetone, then alcohol, and finally, deionized water. Each cleaning process was performed for 15 min. After cleaning, the Si/SiO_2_ substrate was dried using nitrogen gas, followed by a hydrophilic treatment using plasma to enhance surface properties. Then, the cleaned Si/SiO_2_ substrate was then positioned on a heated stage set to maintain a steady temperature of 50 °C. After the temperature had stabilized, 20 μL saturated precursor was carefully added, swiftly followed by the addition of 10 μL CH_2_Cl_2_. Following the evaporation of solutions, a layer of CsPbBr_3_ MPs was successfully formed on the surface of the Si/SiO_2_ substrate.

## 3. Results and Discussion

### 3.1. Morphology and Structure Characteristics

Figure 2 shows the morphologic characteristics of CsPbBr_3_ MPs synthesized by FSA. The large-area SEM image (Figure 2a) confirms that the samples exhibit uniform dispersion across the Si/SiO_2_ substrate, with no evident signs of agglomeration, and the as-fabricated CsPbBr_3_ MPs exhibit uniformly rectangular shapes. Furthermore, the distribution of sizes across the samples is consistently uniform. These observations underscore the reliability and precision of the rapid antisolvent method used for synthesis, ensuring consistent sample production. The individual CsPbBr_3_ MP energy-dispersive spectroscopy (EDS) mappings demonstrate that the distributions of Cs, Pb, and Br are evenly spread throughout the entire MP, as illustrated in Figure 2b. The fabrication processes of CsPbCl_3_ MP include two steps: nucleation and nuclei growth. The nucleation rate can be expressed as the following formula [29]:$$\frac{dN}{dt}=A\exp\left(\frac{-∆G}{k_{b}T}\right)=A\;exp(\frac{16\pi \gamma^{3}V_{m}^{2}}{3k_{b}^{3}T^{3}N_{A}^{2}{\left(\ln S\right)}^{2}}),$$
where *N* is the number of nuclei, *A* is the pre-exponential factor, *k_b_* is the Boltzmann constant, *N_A_* is the Avogadro constant, *S* is the degree of supersaturation, and *T* is the temperature. According to this equation, the main factors affecting the number of nucleation are *T* and *S*. The additive of antisolvent increases *S*, leading to the formation of a large number of crystal nuclei, which grows into MP with the evaporation of precursor solution. The rate of precursor solution is controlled by *T*. When the CB is incorporated into the precursor, its concentration diminishes. As the antisolvent is introduced, the number of nuclei produced will be greatly reduced. This arrangement ensures that during the ensuing growth process, each nucleus has ample space and resources for growth, thereby guaranteeing both its crystal quality and size.

CB was incorporated into our approach to regulating the concentration of the precursor, with two primary motivations behind this decision. First, CB is frequently employed as an antisolvent in perovskite synthesis, and its inclusion is known to facilitate certain enhancements in the surface quality of the perovskite. Second, given that CB possesses a slightly lower boiling point (132 °C) than DMF (153 °C), it prevents the re-dissolution of perovskite following initial evaporation, thereby preserving the final morphology of CsPbBr_3_ MPs.

XPS is used to determine the structural composition of the heterojunction single crystals shown in Figure 3a. In Figure 3b–d, several distinct characteristic peaks exist in the representative XPS spectrum of the CsPbBr_3_ MPs, which should come from the Cs 3d (3/2 and 5/2), Pb 4f (5/2 and 7/2), and Br 3d (3/2 and 5/2) core orbital levels of the CsPbBr_3_ materials [30]. The binding energies for the Cs 3d, Pb 4f, and Br 3d regions are consistent with CsPbBr_3_. In addition, the molar ratio of Cs, Pb, and Br atoms was calculated to be 1:1:3 based on this XPS spectrum, which is in good agreement with the stoichiometric CsPbBr_3_.

To investigate the crystal structure and growth mechanism of directional CsPbBr_3_ MPs, we implemented X-ray diffraction (XRD) and transmission electron microscopy (TEM) characterizations. As shown in Figure 4a, all the CsPbBr_3_ MPs’ sharp diffraction peaks are indexed to the tetragonal perovskite phase of CsPbBr_3_ [31]. Three peaks are resolved at 15.2°, 30.3°, and 30.7°, which index to (001), (002), and (200) planes of the orthorhombic phase of CsPbBr_3_, respectively. Figure 4b shows the high-resolution TEM image and the measured lattice spacings of the CsPbBr_3_ MP is approximately 0.58 nm, corresponding to (001) lattice planes of tetragonal perovskite CsPbBr_3_.

### 3.2. Optical Characteristics

Figure 5a shows the temperature-dependent PL of single CsPbBr_3_ MP from 78 K to 298 K. The PL spectra were asymmetric in shape with a main peak at ~2.33 eV and a shoulder at ~2.37 eV under 298 K. Via Gaussian fitting, two peaks are obtained, which are Peak 1 (high energy) and Peak 2 (low energy), respectively. Based on the position of the peaks and the trend of increasing intensity with temperature, we assume that both peaks are from exciton emission [32]. The variation in peak energy with temperature from 78 K through 298 K for the two peaks is shown in Figure 5b. Thermal expansion and electron–phonon interaction lead to the change in the band gap of the semiconductor with temperature [33]. When the temperature is reduced from 298 K to 200 K, thermal expansion is a dominating factor. Thermal expansion of the lattice tends to increase the interaction between two valence orbitals (s and p), which leads to narrowing the valence bandwidth and, hence, the forbidden gap decreases [34]. In the meantime, as temperature decreases, electron–phonon scattering is lower, leading to the redshift of the emission peak [35]. The change of peak position is due to lattice thermal expansion and electron–phonon coupling. The free excitons are band recombination, and the conduction band and valence band are sensitive to the change in temperature. Figure 5b, both Peak 1 and Peak 2 broaden monotonically with the increasing temperature. The energy difference between Peak 1 and Peak 2 is about 10 meV at 78 K, and as the temperature increases, the energy difference becomes larger, and the energy difference is about 40 meV at 298 K. In addition, Peak 2 shows a significant redshift from 200 K to 298 K, indicating that the electron–phonon coupling is dominant as the temperature increases; that is, the electron–phonon coupling strength is higher than that of the free exciton. Therefore, Peak 2 is likely to be a localized exciton. [36,37]. The temperature-dependent PL study gives information about the physical parameters of CsPbBr_3_ such as exciton binding energy (*E_b_*), longitudinal optical phonon energy (*E_ph_*), inhomogeneous broadening (*Γ*_0_), and exciton–phonon coupling strength (*Γ_op_*). As expected, PL intensity for all peaks decreases with increasing temperature. The PL intensity as a function of temperature is plotted and fitted using an Arrhenius equation [38]:$$I(T)=\frac{I_{0}}{1+Aexp[-E_{b}/k_{b}T]}$$
where *I*_0_ is the PL intensity at 0 K, *E_b_* is the activation energy, and *k_b_* is the Boltzmann constant. Here, the exciton binding energy *E_b_* of Peak 1 can be fitted as 45.9 meV. These values were very close to the binding energy (40 meV) previously reported for CsPbBr_3_, indicating the significant contribution of thermal dissociation of excitons to the decrease in PL intensities [39]. Assessing the temperature-dependent emission broadening in Figure 5d,e, the intrinsic FWHM (*Γ*_0_) of both peaks grows with increasing temperatures. The Peak 1 emission broadens faster than Peak 2 with rising temperature, suggesting enhanced carrier scattering in these bands [40,41]. The temperature-dependent excitonic line width of band-to-band transitions within semiconductors is relatively well understood, being described by a Boson model [42]:$$\Gamma \left(T\right)=\Gamma_{0}+\sigma T+\Gamma_{op}/\left(exp\left(E_{op}/k_{b}T\right)-1\right)$$

In this equation, the three terms from left to right denote the contributions of inhomogeneous broadening, acoustic phonons, and optical phonons, respectively, with *σ* the exciton–acoustic phonon interaction coefficient, *Γ_op_* the exciton–optical phonon coupling coefficient, and *E_op_* the average optical phonon energy. The fitted values of *Γ*_0_, *σ*, *Γ_op_*, and *E_op_* are summarized in Table 1.

This shows that the linewidth of the two peaks mainly comes from the exciton–phonon interaction.

### 3.3. Lasing

To obtain the laser characteristics of the sample, we used a 400 nm, 1 kHz femtosecond laser as the pump source to excite an individual CsPbBr_3_ MP, and all tests were performed at room temperature. The evolution of the PL spectral profile with the pump fluence is shown in Figure 6a. The PL spectrum is characterized by a broad spontaneous emission band centre at ~535 nm with the FWHM of ~20 nm when the CsPbBr_3_ MP is pumped at low pump fluences. When the pump fluence is increased up to 39.5 μJ cm^−2^, a new narrow peak begins to emerge at ~540 nm. When the pump fluence is further increased, the new-emerging narrow peak centre at ~540 nm becomes dominant over the spontaneous emission background. The peak is well-fitted by a Gauss function with an FWHM of 0.6 nm. The *Q* factor was calculated to be ≈ 913.3 using the equation *Q* = *λ*/Δ*λ*, where *λ* is the peak centre wavelength and Δλ is the peak width. Figure 6a inset that bright green emission from the rod ends was observed under an excitation density higher than *P*_th_. In a square WGM resonator, the mode spacing Δ*λ* at wavelength *λ* is given by [43,44]:$$∆\lambda =\frac{\lambda^{2}}{2\surd 2n_{g}L}$$
where *n_g_* is the group refractive index. The mode spacing Δ*λ* between two adjacent modes decreases with the increase in the edge length *L*. When the excitation fluence surpasses the lasing threshold, spontaneous emission is selected and restricted by the WGM cavity. This restriction causes a large amount of photons to only be released from the edges of the square cavity [45]. As a result, the micro-pillar edge, and especially the corners, exhibit significantly stronger emissions [11]. The CsPbBr_3_ Mott density (1.8 × 10^17^ to 4.7 × 10^17^ cm^−3^), results in the electron–hole plasma, which is also its gain characteristic. In our report, the lasing threshold was derived at ~34.1 μJ cm^−2^ and then the carrier density at the threshold of the CsPbBr_3_ MP was estimated to be 6.2 × 10^19^ cm^−3^. Therefore, this is the main cause of the redshift of the laser mode [46]. Figure 6b plots the PL intensity and FWHM as a function of the pumping density. An inflection point can be observed during the pumping density increase procedure, confirming the evolution from spontaneous emission to stimulated emission at ≈34.1 μJ cm^−2^. At this point, the FWHM will also dramatically decrease. Figure 6c is a statistical diagram of the threshold of random CsPbBr_3_ MP. The threshold of the sample varies from 30–60 μJ cm^−2^, which is attributed to the difference in size of CsPbBr_3_ MP [47,48]. 

### 3.4. Stability

Long-term working stability is significantly important for practical application. As shown in Figure 7a, the CsPbBr_3_ MP shows excellent optical stability with the PL intensity sustained for more than 180 min, corresponding to 1.08 × 10^7^ laser shots. The PL intensity measured at 0 and 180 min shows almost identical. Interestingly, the PL intensity of CsPbBr_3_ MP tended to increase first with the excitation density, possibly due to humidity-induced self-healing of the perovskite lattice or the photocuring phenomenon [49,50]. The CsPbBr_3_ MP without any encapsulation retained 95% of its original PL in ambient conditions for 12 weeks (Figure 7b). We believe the smaller specific surface area and the small number of defects in the CsPbBr_3_ MP are the key points that make it more stable than other nanostructured perovskite materials. The above results demonstrate that the large CsPbBr_3_ MPs have great potential for practical applications in nonlinear optical devices.

## 4. Conclusions

In this study, we successfully synthesized monodispersed CsPbBr_3_ MPs on Si/SiO_2_ substrate by optimizing the rapid antisolvent technique, paying particular attention to understanding the role of CB in the process. Temperature-dependent PL analysis elucidated that the exciton binding energy and linewidth are predominantly governed by exciton-photon interactions. Furthermore, we achieved a WGM laser with a threshold of 34.1 μJ cm^−2^ at room temperature. Remarkably, the PL intensity remained largely unaltered after 12 weeks of environmental exposure. These findings collectively suggest that CsPbBr_3_ MPs prepared by this method possess significant potential for applications in integrated photonics and optoelectronic devices.

## Figures and Tables

**Figure 1 nanomaterials-13-02116-f001:**
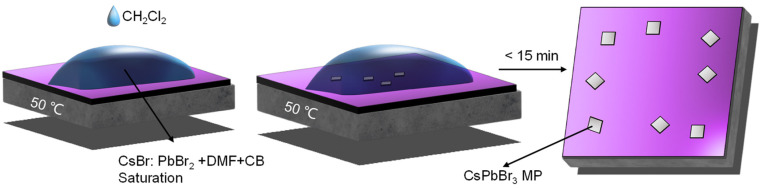
Schematic diagram of the CsPbBr_3_ MP.

**Figure 2 nanomaterials-13-02116-f002:**
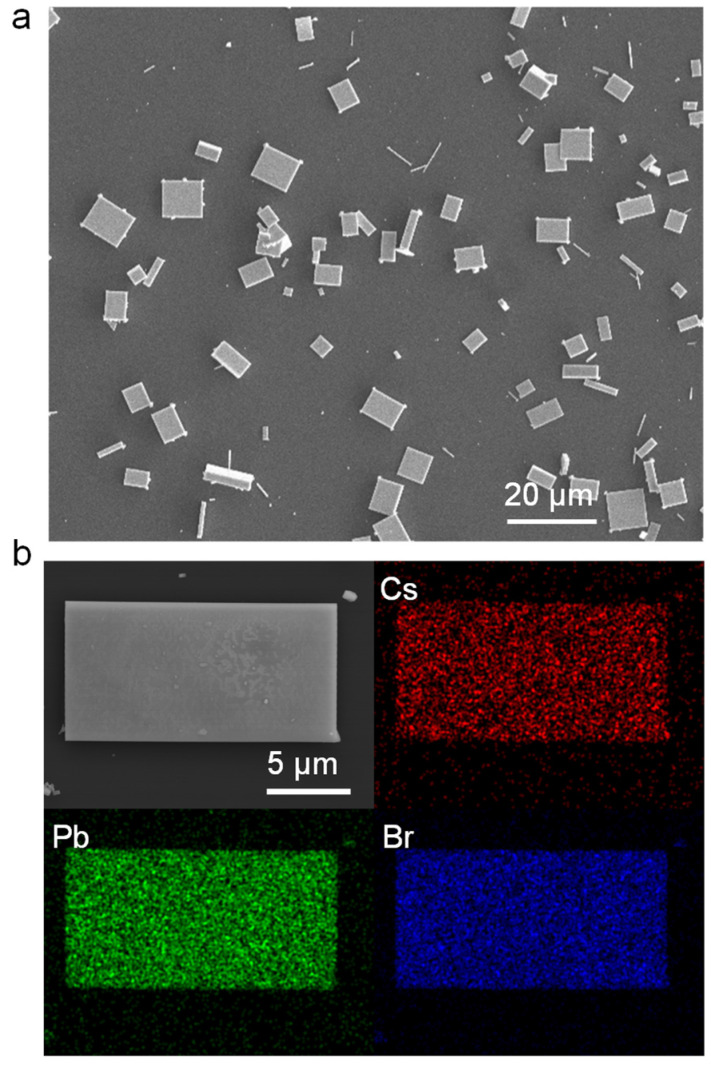
Morphology characterizations of FAS-fabricated CsPbBr_3_MPs. (**a**) Large-area SEM image. (**b**) SEM image of single CsPbBr_3_ MP and the corresponding elemental mapping of Cs, Pb, and Br, respectively.

**Figure 3 nanomaterials-13-02116-f003:**
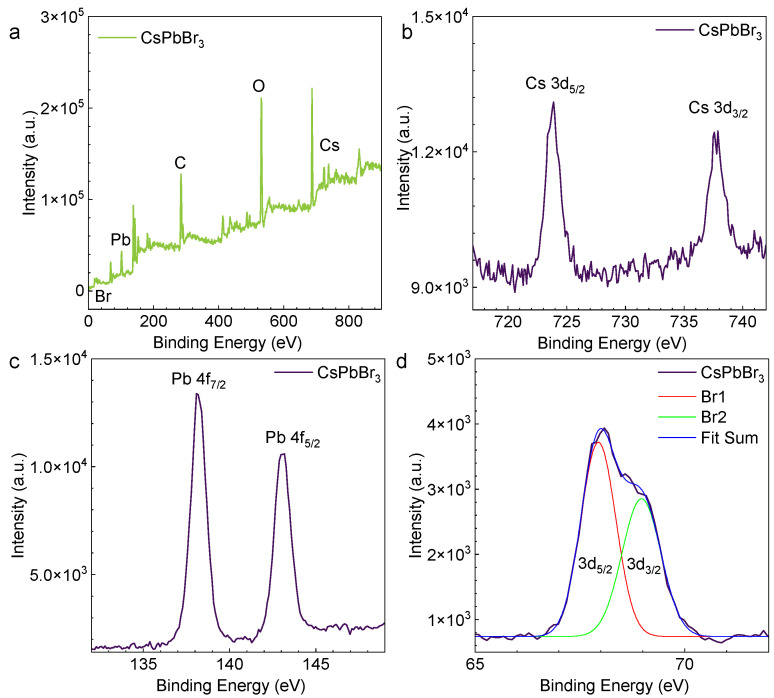
The XPS spectra of FAS-fabricated CsPbBr_3_ MPs. (**a**) XPS characterization of the CsPbBr_3_ MPs. (**b**–**d**) Corresponding XPS spectra of Cs 3d, Pb 4f, and Br 3d showing identical chemical bonding.

**Figure 4 nanomaterials-13-02116-f004:**
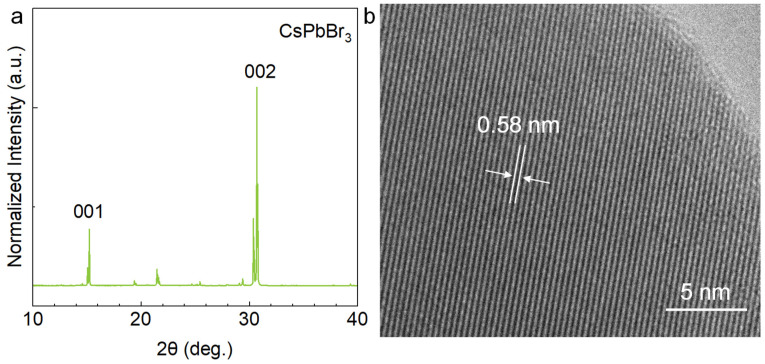
Structure characterizations of FAS-fabricated CsPbBr_3_ MPs. (**a**) XRD pattern. (**b**) The high-resolution TEM image.

**Figure 5 nanomaterials-13-02116-f005:**
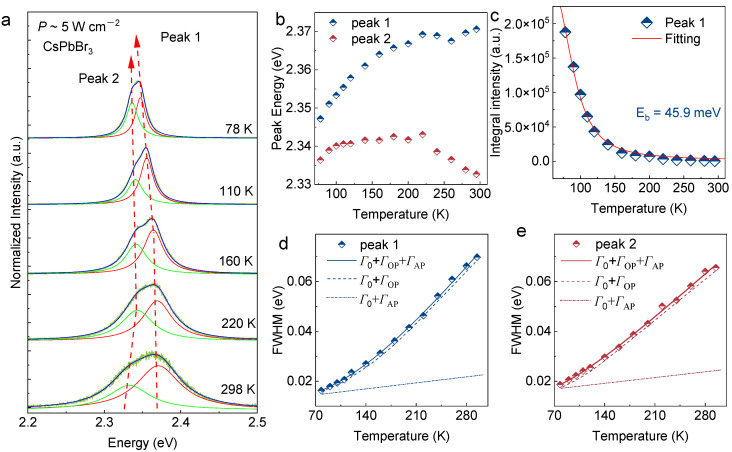
Temperature-dependent PL spectra of FAS-fabricated CsPbBr_3_MP. (**a**) Temperature-dependent PL spectra of single CsPbBr_3_ MP in the temperature range of 78–298 K (CW laser 405 nm). (**b**) Emission energy of two PL emission peaks. (**c**) Integral PL intensities of Peak 1 specimens recorded at low temperatures. FWHM of PL bands recorded from (**d**) Peak 1, and (**e**) Peak 2 specimens at low temperatures.

**Figure 6 nanomaterials-13-02116-f006:**
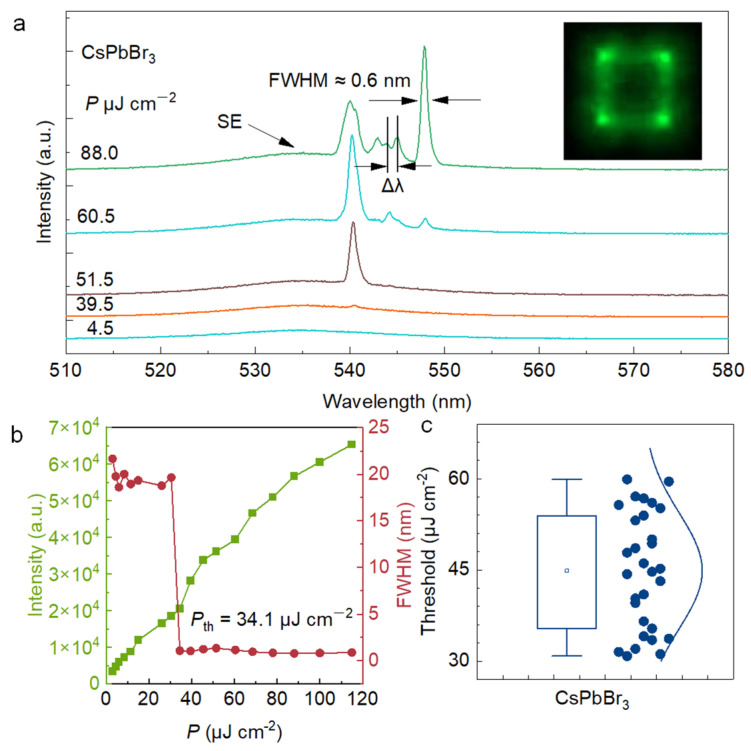
Lasing of single CsPbBr_3_MPs. (**a**) PL spectra of a FAS-fabricated CsPbBr_3_ MP at pump densities of 4.5, 39.5, 51.5, 60.5, and 88.0 μJ cm^−2^, respectively. Inset: PL emission image above threshold. Length of MP side ≈ 10 μm. (**b**) Integrated PL intensity and FWHM as a function of pump density ranging from 3.0 to 115.0 μJ cm^−2^. The threshold is ≈34.1 μJ cm^−2^. (**c**) Threshold statistics of FAS-fabricated CsPbBr_3_ MPs.

**Figure 7 nanomaterials-13-02116-f007:**
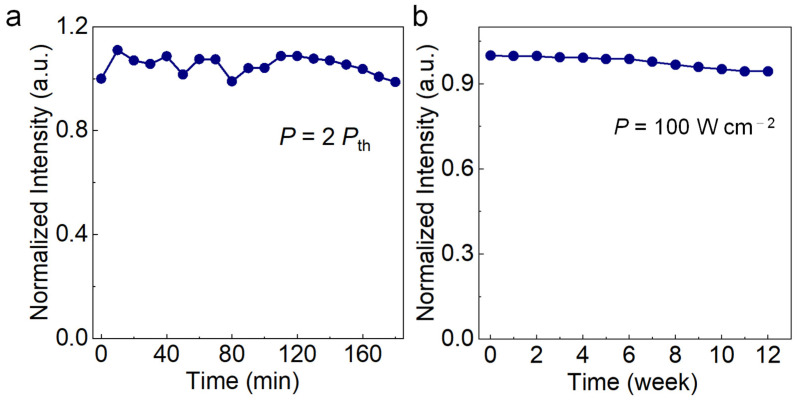
Stability of FAS-fabricated CsPbBr_3_ MPs. (**a**) Normalized PL intensity of the CsPbBr_3_ MP under pulsed laser excitation at 2 *P*_th_. (**b**) Normalized PL stability of the CsPbBr_3_ MP at room temperature under 405 nm CW lasing.

**Table 1 nanomaterials-13-02116-t001:** The fitted parameters of Peak 1 and Peak 2.

	*Γ*_0_ (meV)	σ (meV/T)	*Γ_op_* (meV)	*E_op_* (meV)
Peak 1	7.88	0.104	58.8	41.7
Peak 2	14.4	0.158	60.5	20.0

## Data Availability

All data are available in the manuscript.

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
