# Peer review of "Ultrafast Antisolvent Growth of Single-Crystal CsPbBr3 Microcavity for Whispering-Gallery-Mode Lasing"

_nanomaterials, 2023, doi:10.3390/nano13142116_

Round 1
Reviewer 1 Report
The manuscript reports fabrication of CsPbBr3 single crystalline microplates investigation of their optical properties and demonstration of lasing. The major focus is on lasing. Authors clearly demonstrate formation of a narrow lasing line and investigate its properties. This demonstration may be useful for some nonlinear optical applications. The manuscript is quite well prepared, majority of conclusions are clear. However, several corrections and clarifications should be made before publication.
1. Ln 180. Fig. 2a should be Fig. 5a.
2. Ln. 183 “Based on the position of the peaks 183 and the trend of increasing intensity with temperature, we assume that both peaks are from excitation emission[32].”
The sentence is very unclear. What is ” increasing intensity with temperature” when Fig.2c clearly demonstrates strongly decreasing intensity.
“both peaks are from excitation emission” what else could cause the peaks.
3. Fig. 5a demonstrates that separation between two peaks increases with temperature. Authors talk only about bandwidth, some comment about separation would be useful.
4. Ln 189. “Thermal expansion of the lattice tends to decrease the interaction between two valence orbitals (s and p), which leads to narrowing the valence bandwidth 190 and hence the forbidden gap increases[34].” But Fig. 5a, in my opinion, demonstrates the opposite, the luminescence band shifts to the higher energy side with temperature.
5. Ln. 234 “ The spectral linewidth of the narrow PL peak at ~540 nm.” The sentence is grammatically incorrect, and I don’t think it is necessary.
6. Figure 6a. What is the origin of the band at about 548? Authors discuss about the bandwidth of the 540 nm band but in Figure 6a they show bandwidth of the 548 nm band.
Minor English corrections are necessary.
Reviewer 2 Report
The paper of Li Zhang, Xin Xin Li, Yi Meng Song and Bingsuo Zou, "Ultrafast Antisolvent Growth of Single-Crystal CsPbBr3 Micro-cavity for Whispering Gallery Mode Lasing" is devoted to the technological development of CsPbBr3 perovskites. The topic of the article seems to be important. In principle, the authors' statements do not contradict the laws of nature and, therefore, the article can be published. However, while reading the article, I had a lot of comments.
It is not very clear what the novelty of the work is. The authors claim that the novelty is that they only added chlorobenzene growing these structures. If this is all the novelty, it is not enough to publish the paper. Perhaps there are more. The authors should describe the novelty in more detail.
On page 6, the reference is to Figure 2. However, the context shows that Figure 5 is meant.
Very often authors miss the space between the text and the references[X] .
Lines 183 -184
“Based on the position of the peaks and the trend of increasing intensity with temperature, we assume that both peaks are from excitation emission[32]”. What is "excitation emission"? What other kinds of emission are there? This sentence makes no sense at all. Maybe the authors meant exciton emission?
Lines 185 – 187
“The variation in peak energy with temperature from 78 K through 298 K for the two peaks are shown in Figure 5b. thermal expansion and electron–phonon interaction is lead to the change in the forbidden gap of semiconductor with temperature.” Generally, it is believed that the band gap width decreases with increasing temperature in accordance with the formula of Y.P.Varshni. According the authors, the bandgap width increases with increasing temperature, especially for peak 1 in Figure 5. The electron-phonon interaction leads to a change in the exciton distribution function in the exciton band (for free excitons) thereby leading to a broadening of the exciton emission line. However, the electron-phonon interaction cannot lead to the blue shift of the free exciton emission line. The blue shift with increasing temperature caused by electron-phonon interaction can in principle appear for localized excitons, due to temperature delocalization, which seems to be observed by the authors. The authors should describe this one in more detail.
Lines 191 – 192
“At the meantime, as temperature decreases, electron–phonon scattering is lower, leading to the blueshift of the emission peak.” From the context, we can conclude that authores are talking about the luminescence of free excitons. Since localized excitons cannot scatter. Thus, the whole paragraph is erroneous.
Lines 226 -235
At weak excitation photoluminescence was observed at a wavelength of 537 nm. Under strong excitation, emission was observed at a wavelength of 550 nm. It is not clear what became of the exciton emission and what emits under strong excitation? From the edge of the sample there is always more intense emission due to waveguide modes. So, it is not necessary to involve WGM. But where did the radiation from the center of the sample go? The authors should clarify this in more detail.
Lines 238 -239
"In a square WGM resonator, the mode spacing Δλ at wavelength λ is given by[41,42]: It is not clear which mods we are talking about. Where are they in picture 6?
I believe that the article should be significantly refined.
Round 2
Reviewer 2 Report
Despite the fact that I still have questions about the article. In particular, concerning the blue shift of the emission line with increasing temperature and the red shift with increasing intensity of optical excitation, I believe that in the presented form the article can be accepted for publication.